# Differences in Multifocal Electroretinogram Study in Two Populations of Type 1 and Type 2 Diabetes Mellitus Patients without Diabetic Retinopathy

**DOI:** 10.3390/jcm11195824

**Published:** 2022-09-30

**Authors:** Pedro Romero-Aroca, Raul Navarro-Gil, Gibet Benejam, Montse Vizcarro, Marc Baget-Bernaldiz

**Affiliations:** 1Ophthalmology Service, University Hospital Sant Joan Reus, 43204 Reus, Spain; 2Institut de Investigacio Sanitaria Pere Virgili (IISPV), Universitat Rovira & Virgili, 43204 Reus, Spain

**Keywords:** multifocal electroretinogram, type 1 diabetes mellitus, type 2 diabetes mellitus, diabetic retinopathy, retina neuropathy

## Abstract

(1) Background: Diabetic retinopathy (DR) is a diabetes mellitus (DM) complication where neurodegeneration plays a significant role. The aim of our study was to determine the differences between type 1 DM (T1DM) and 2 DM (T2DM) in the multifocal electroretinogram (mERG).; (2) Methods: A mERG study was performed in two groups, a T1DM group with 72 eyes of 36 patients compared with 72 eyes of 36 patients with T2DM, randomly selected from our DM databases, without DR. We studied how HbA1c and DM duration affects amplitude and implicit time of mERG; (3) Results: the study of DM duration shows patients with T1DM have lower amplitude values compared to T2DM patients, although implicit time increases in patients with T2DM. HbA1c over 7% only affects T1DM patients with an increase of implicit time; (4) Conclusions: the retinas of patients with T1DM seem more sensitive to changes in HbA1c levels than in patients with DMT2, although the duration of diabetes affects both types of DM patients.

## 1. Introduction

Diabetic retinopathy (DR) is a neurovascular complication in patients with both type 1 (T1DM) and type 2 (T2DM) diabetes mellitus (DM) that can lead to reduced or complete loss of vision. If detected promptly, the risk of vision loss can be reduced by 95%, but the disease is asymptomatic in its early stages [1].

Globally, DR is the fifth leading cause of blindness and visual impairment and the most common in working-age adults. In Europe, 3–4% of people have DM, although, more importantly, up to 20% of diabetics remain undiagnosed. In addition, due to the increasing prevalence of diabetes, associated diseases such as DR are expected to increase significantly in the coming years [2].

DR is often not diagnosed until the first symptoms of vision loss appear. Poorly controlled blood glucose levels have a great impact on the development of DR. Patients with T1DM spend many more years of their lives at risk of developing DR than patients with T2DM, since they are exposed to variations in glycemia levels for more time. However, DR has a much greater impact on T2DM, probably because the time of diagnosis of DM does not coincide with its onset. The patient may have spent several years with high blood glucose levels in a phase of prediabetes or unknown diabetes for a long time. This will make the detection of DR in patients with T2DM more frequent [3]. Currently, DR is a complex condition in which neurodegeneration plays a significant role [4,5]. In fact, the American Diabetes Association (ADA) defines DR as a highly tissue-specific neurovascular complication [6]. We can differentiate two stages of DR: the early stage and advanced stage. The early stage comprises non-apparent DR (no microaneurysms nor haemorrhages), mild DR (only microaneurysms) and moderate DR (microaneurysms plus haemorrhages or hard exudates). The advanced stage includes vision-threatening conditions such as severe DR, proliferative DR and diabetic macular edema (DME).

In the pathogenesis of DR, genes have been found that can determine its appearance by mediating through up-regulation of the VEGF molecule. The presence of elevated VEGF in the retina induces the appearance of retinopathy due to alteration of the blood–retinal barrier and inducing ischemia [7,8].

For DM, we can use a multifocal electroretinogram (mERG) to detect the neuropathy prior to vasculopathy (microaneurysm, haemorrhages etc.). In many studies, mERG alterations prelude DR vasculopathy. In a study conducted by the European Consortium for the Early Treatment of Diabetic Retinopathy (EUROCONDOR), a sample of 442 T2DM patients yielded 58% of patients with mERG abnormalities in the absence of any visible retinopathy. The study concluded that neurodegeneration plays a role in the pathogenesis of DR in many, but not all, patients with T2DM [9].

Also, some studies have associated changes in ERGm after intravitreal anti-VEGF therapy [10]. Thus, in a previous study we have shown that the risk factors for diabetic retinopathy affect T1DM and T2DM differently [11]. Given this background, the aim of the present study is to determine any differences in the parameters detected by mERG.

## 2. Materials and Methods

### 2.1. Study Design

A cross-sectional study was conducted from 1 January 2019 to 31 December 2019.

A mERG study was carried out on two groups of Caucasian patients. The T1DM group included a sample of 72 eyes of 36 patients from our database. The T2DM group included the same sample size from our database. The two groups were matched according to sex, duration of diabetes mellitus, and in the case of patients with T2DM, only those who had insulin, or insulin and oral antidiabetics, for DM treatment were included, so that the groups were in effect homogeneous.

### 2.2. Ethical Adherence

This study was performed in adherence to local legal requirements [local ethics committee of Hospital Universitari Sant Joan de Reus, approval no. 11-05-26/proj5], in accordance with the revised guidelines of the Declaration of Helsinki. Informed consent was obtained from all participants in the study.

### 2.3. Power of the Study

We estimated the detection of a 95% increase in risk with an accuracy interval of 3%. The calculations were based on the consideration of variables involving two errors in their determination.

Amplitude of the SOK P1 wave: This has been considered an error of 20 nanovolts per degree squared [nV/deg2] in its determination for every ring studied, due to possible technical mistakes.Errors in measuring the thickness of the retina with optical coherence tomography [OCT], with a sampling error of 5 mm, due to possible failure of the technique used.

Inclusion criteria:

Type 1 and 2 DM patients with no diabetic retinopathy.

Exclusion criteria:Patients with other types of DM;Patients with cataracts or other opacities;Patients with glaucoma or previous ocular surgery;Patients with myopia > 6 diopters; Patients with macular pathology; Patients with previous nephropathy, stroke or myocardial infarction.

### 2.4. Multifocal-Electroretinogram

The mERG were carried out with a RETIscan^®^ (ROLAND INSTRUMENTS, Wiesbaden, Germany), following the recommendations of the International Society for Clinical Electrophysiology Vision (ISCEV) in 2011 [12]. The patient was placed in front of an LCD 19 monitor, onto which we projected a hexagonal matrix of 61 flicker lights, transitioning from white to dark at high frequency (75 Hz). This pattern of stimulation was performed under photopic conditions in order to achieve electrophysiological responses in the cone and bipolar cells.

The mfERG includes a first negative wave, or N1 (which is equivalent to an a-wave in classic ERG), followed by a positive P1 wave. The set of N1-P1 biphasic waveforms is known as the second order kernel (SOK) or second order response

The two electrophysiological variables selected to assess the electrophysiological responses at the fovea (R1), perifovea (R2) and parafovea (R3) were the amplitude (A) and the implicit time (IT) of the P1 wave of the second-order kernel (SOK) (Figure 1).

The amplitude was achieved per unit of area, or the ring ratios measured in nanovolts per degree squared (nV/deg2), and the amplitude was quantified in each ring. IT is the time to reach the maximum amplitude in every macular region studied.

### 2.5. Statistical Methods

Data analysis used the SPSS software package, version 22.0 (IBM^®^ Statistics, Chicago, IL, USA). The independent variables included the amplitude and implicit time of the P1 wave in the first three rings. The dependent variables included age, sex, arterial hypertension, DM duration and HbA1c level.

In the second step, the patients were grouped according to two risk factors as variables: a DM duration of <15 years or ≥15 years, and HbA1c of <7% or ≥7%. T, the cutoff of DM duration at 15 years, is due to the fact that in this figure it is considered that DR begins to appear in more than 50% of patients [11]. The cutoff of HbA1c at 7% is due to the fact that it is considered good control if the HbA1c is in the value of 7% in most diabetic patients. 

A descriptive statistical analysis of the quantitative data was carried out, and for qualitative data we used the analysis of frequency and percentage in each category. Differences between variables included in the analyses were evaluated using Student’s *t*-test to compare two variables, or one-way ANOVA when comparing more than two variables. In cases of three or more samples, to determine which individual samples were different from one another, we used post-hoc tests. The inferential analysis for qualitative data used the chi-square table and the determination of McNemar’s test for categorical data, or Cochran’s Q-test when there were more than two fields. The hypothesis of normality in the quantitative variables was tested using the Kolmogorov–Smirnov test.

## 3. Results

### 3.1. Demography of the Sample

One group included a sample of 72 eyes of 36 patients, randomly selected from our database of patients with T1DM. The other group was of the same sample size, randomly selected from our database patients with T2DM. The two groups were matched according to sex, duration of diabetes mellitus, and in the case of patients with T2DM, only those who had insulin, or insulin and oral antidiabetics, for DM treatment were included so that the groups were in effect homogeneous. 

Table 1 presents the differences between groups by demographic data. There were no differences according to sex, current age, DM duration or HbA1c. Differences in the prevalence of arterial hypertension was normal due to its greater frequency in T2DM.

With respect to the second step of the analysis (the differences between groups according to 7% cut-off in HbA1c and 15 years cut-off in DM duration) we observed a difference for patients with HbA1c of >7%, which is more frequent in T2DM, at a significance of *p* < 0.001.

### 3.2. Study of the mERG Variables in Each Group of Type 1 and Type 2 Diabetic Patients

The study was carried out in the three central rings. We ignored the results of the fourth and fifth rings as any changes have been shown in previous studies not to be significant for any of the variables studied [10,11,12].

Table 2 gives the mean values of amplitude an implicit time of mERG in the first three central rings of the macula. We found that differences in amplitude were significant between the two groups. Mean amplitudes were lower in T1DM. There were no differences in implicit time values.

#### 3.2.1. Study of the mERG Result According to HbA1c Levels

Table 3 gives the results of the two groups according to an HbA1c level of below or above 7%. In T1DM patients, amplitude decreased significantly in the first ring but not in the other two rings, and in implicit time all three rings increased significantly. In T2DM patients, none of the values of amplitude or implicit time were significant in the three rings if hbA1c values increased above 7%.

#### 3.2.2. Study of the mERG Result According to DM Duration

Table 4 shows the results of amplitude and implicit time in the two groups according to DM. In the T1DM group there were no differences in the amplitude or implicit time in the three rings studied. In the T2DM group, we found a significant decrease in the amplitude and a significant increase in implicit time in all rings.

## 4. Discussion

The effect of diabetes mellitus (DM) on the retina has been studied through mERG for many years, in patients with and without diabetic retinopathy (DR). Alterations in the parameters of the mERG (amplitude and implicit time) in patients with DR has led to the assumption that there is a diabetic neuropathy in the retina that might appear prior to DR [13,14,15]. Some publications have attempted to show that alterations in the mERG could be used as predictors of the later development of DR [16,17]. Likewise, there are publications that report improvements in mERG parameters after treatment of DR [18,19].

The aim of the current study was to compare mERG amplitude and implicit time values in T1DM and T2DM patients without DR. There are various publications that have studied such changes in the mERG parameters in patients with DM, but they either did not specify type or studied only patients with T2DM. There are a few studies that have compared both forms of diabetes mellitus, such as Bronson-Castain et al. [20], but here only adolescent patients with both types of DM were studied.

In the first part of the study, we compared the results of amplitude and implicit time in the three central rings between both DM groups, and the results show that patients with T1DM have lower amplitude values than patients with T2DM. Although this may be due to the affectation of DM, it can also be secondary to the current age of T1DM patients of the sample; in the current study, the mean age was 39.75 ± 10.31 years, an age that we believe to be high for patients with T1DM in the studies that we have published [21]. These differences might affect the mERG parameters, especially amplitude, as Tzekoz et al. [22] showed that age is one of the factors that can influence the decrease in amplitude in the mERG. Likewise, patients with T1DM who are older also have a longer DM duration.

The two variables (HbA1c and DM duration) that we focused on are the most important risk factors in the development of DR [23,24]. Although other factors are also important, we did not take them into account in this study because of their varying prevalence in the two types of DM. For example, the number of T1DM patients affected by arterial hypertension in our study sample was only 11.6% (4 patients), compared to 88.4% (30 patients) of T2DM, which can cause errors when analyzing results. Additionally, another variables could be renal complications.

One focus of the present study was on the effect of DM duration of less than or greater than 15 years. Our results show that duration does not affect amplitude in patients with T1DM but decreases it in T2DM. This could be explained by T1DM patients having amplitude values lower than T2DM patients, as shown in Table 2, and having longer DM duration. Srinnivasan et al. [25] studied a series of 85 patients with T2DM who had a DM duration of greater than 10 years and without DR. They reported a decrease in amplitude and an increase in implicit time compared to patients with DM duration of less than 10 years.

High levels of HbA1c affect patients with T1DM by decreasing amplitude and increasing implicit time, while neither of these two variables are affected in patients with T2DM. The absence of alterations in mERG results in patients with T2DM is consistent with the studies of Srinivasan et al. [25] and Santos et al. [9], who studied 449 patients with T2DM recruited from the Open Consortium for the Early Treatment of Diabetic Retinopathy (EUROCONDOR). It is important, therefore, that our results for patients with T1DM show that high levels of HbA1c influence the mERG scores.

We found that changes in blood glucose, measured by HbA1c, affect patients with T1DM, similar to other studies. Klemp et al. [26] reported that hyperglycaemia induced a delay in the first- and second-order kernel implicit times in patients with T1DM and without DR. Likewise, Tyberg et al. [27] reported that patients with insulin-dependent diabetes (T1DM) have specific abnormalities in both the first- and the second-order component of the mERG. 

Our results support the hypothesis that hyperglycaemia changes the metabolism of the retina, probably inducing changes in inner retinal function, including in T1DM patients without DR. One limitation of this study is the small sample size, with only 72 patients, despite being randomly selected from our databases. A bigger study sample would strengthen the results, as would the inclusion of other risk factors in addition to DM duration and HbA1c. An important strength of this study includes the random selection of patients according to sex, DM duration and insulin treatment from a DM database representative of our population.

Finally, in a changing world that has experienced the COVID-19 pandemic, the chance of finding new systems for the early detection of prevalent pathologies such as DR is very important, especially by means of telemedicine [28,29]. This is the case with ERGm, which can be read by ophthalmologists from remote sites.

## 5. Conclusions

In patients with type 1 diabetes mellitus, the retina is functionally more sensitive to changes in blood glucose than in patients with type 2 diabetes mellitus. On the contrary, DM duration affects retinal functionality more in both types of diabetes studied. 

## Figures and Tables

**Figure 1 jcm-11-05824-f001:**
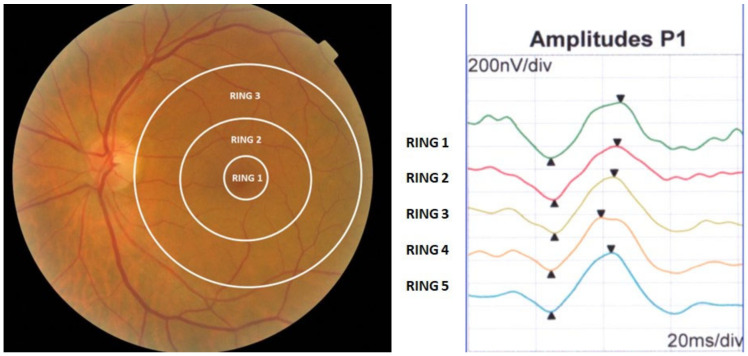
Diagram of the three central rings studied. On the right, the position of the three central rings of the mERG with respect to the eye fundus. On the left, the waves of the five rings of the ERGm, of which we only used the three central ones. This image shows the implicit time (triangle with the vertex up) and the amplitude (triangle with the vertex down).

**Table 1 jcm-11-05824-t001:** Demographic results for both groups of DM patients. * Student’s *t*-test, ** Chi-squared test.

Variable	Type 1 DM	Type 2 DM	Significance (*p*)
Age (mean)	39.75 ± 10.31	64.07 ± 6.06	<0.001 *
Sex (male)	20 patients (55.6%)	21 patients (58.3%)	0.174 **
Arterial hypertension	4 patients (11.3%)	30 patients (84.7%)	<0.001 **
DM duration mean	18.13 ± 8.96	19.81 ± 10.32	0.06 *
HbA1c mean	7.94 ± 1.68	7.94 ± 1.08	0.98 *
Diabetes duration over 15 years	18 patients (51.4%)	20 patients (55.6%)	0.616 *
HbA1c > 7%	16 patients (44.4%)	27 patients (60.4%)	<0.001 *

**Table 2 jcm-11-05824-t002:** Values of mERG parameters studied in both groups. * Student’s *t*-test.

Parameter	T1DM	T2DM	Significance *p* *
Amplitude 1st ring	1.14 ± 0.64	1.87 ± 0.75	<0.001
Amplitude 2nd ring	0.89 ± 0.48	1.67 ± 0.66	<0.001
Amplitude 3rd ring	0.76 ± 0.49	1.56 ± 0.66	<0.001
Implicit time 1st ring	46.28 ± 6.01	47.25 ± 3.61	0.32
Implicit time 2nd ring	47.48 ± 3.29	46.96 ± 2.99	0.33
Implicit time 3rd ring	46.57 ± 3.91	46.12 ± 3.51	0.46

**Table 3 jcm-11-05824-t003:** Values and significance (*p*) of amplitude and implicit time according to HbA1c values. * Student’s *t*-test.

DM Type	Parameter	HbA1c < 7%	HbA1c ≥ 7%	Significance *p* *
T1DM	Amplitude 1st ring	1.28 ± 0.69	0.97 ± 0.54	0.03
	Amplitude 2nd ring	0.99 ± 0.52	0.67 ± 0.66	0.23
	Amplitude 3rd ring	0.87 ± 0.53	0.61 ± 0.41	0.32
	Implicit time 1st ring	44.56 ± 6.02	47.88 ± 6.67	0.02
	Implicit time 2nd ring	45.88 ± 3.16	49.47 ± 2.19	0.01
	Implicit time 3rd ring	44.64 ± 3.95	49.51 ± 1.67	0.02
T2DM	Amplitude 1st ring	1.92 ± 0.82	1.85 ± 0.74	0.44
	Amplitude 2nd ring	1.69 ± 0.75	1.67 ± 0.64	0.22
	Amplitude 3rd ring	1.61 ± 0.73	1.55 ± 0.64	0.27
	Implicit time 1st ring	45.45 ± 4.61	47.81 ± 3.08	0.46
	Implicit time 2nd ring	45.35 ± 2.62	47.46 ± 2.94	0.58
	Implicit time 3rd ring	44.92 ± 2.82	46.49 ± 3.64	0.36

**Table 4 jcm-11-05824-t004:** Values and significance of DM duration and type of DM. * Student’s *t*-test.

DM Type	Parameter	Duration < 15 Years	Duration ≥ 15 Years	Significance *p* *
T1DM	Amplitude 1st ring	1.28 ± 0.67	1.01 ± 0.59	0.79
	Amplitude 2nd ring	1.01 ± 0.53	0.78 ± 0.41	0.06
	Amplitude 3rd ring	0.91 ± 0.52	0.62 ± 0.43	0.18
	Implicit time 1st ring	46.11 ± 4.93	47.05 ± 4.67	0.72
	Implicit time 2nd ring	46.86 ± 3.13	48.06 ± 3.13	0.26
	Implicit time 3rd ring	45.93 ± 4.66	47.62 ± 3.01	0.19
T2DM	Amplitude 1st ring	2.03 ± 0.68	1.73 ± 0.79	0.02
	Amplitude 2nd ring	1.82 ± 0.55	1.56 ± 0.73	0.04
	Amplitude 3rd ring	1.71 ± 0.52	1.45 ± 0.73	0.01
	Implicit time 1st ring	46.48 ± 3.93	47.86 ± 3.24	0.04
	Implicit time 2nd ring	45.83 ± 3.18	47.88 ± 2.51	0.03
	Implicit time 3rd ring	44.69 ± 3.01	47.26 ± 3.51	0.01

## Data Availability

Not applicable.

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
