# Peer review of "Differences in Multifocal Electroretinogram Study in Two Populations of Type 1 and Type 2 Diabetes Mellitus Patients without Diabetic Retinopathy"

_jcm, 2022, doi:10.3390/jcm11195824_

Round 1

Reviewer 1 Report

Pedro Romero-Aroca and co-workers provided a manuscript focused on an interesting issue regarding the differences in retinal alterations between type 1 Diabetes Mellitus (T1DM) and T2DM patients through a study based on a multifocal electroretinogram (mERG) evaluation. The authors highlighted that differences in amplitude of the P1 wave correlate significantly between the two groups of T1DM and T2DM patients with lower mean amplitudes in T1DM and without differences in implicit time values. The scientific approach has been conducted appropriately, reaching the goals claimed at the beginning of the manuscript.  However, the manuscript could be improved by addressing some points

Major points:

 the Introduction section need to be improved by checking and reformulating different sentences. In particular, two sentences from line 33 to line 37 need to be clarify and bibliographic entries should be added.  In line 48 did the authors want to write “preclude or prelude”? Some general sentences and relatives additional bibliographic entries (such as: PMID: 33065984; 25956512; 32685397) should be added to briefly mention the pathogenesis and genetic implications of DR as well as mostly the potential applications of mERG in the management of the pathology.

- The study seems to be lacking regarding the statistical evaluation of the data. Unlike the authors stated in the statistical methods section (line 125 – 127) the Student's t test is used to compare the means between two groups, whereas ANOVA is used to compare the means among three or more groups but when using one independent variable. Probably the authors used groups with variables. This issue need to be checked. Did the authors checked whether their data are characterized by a normal distribution?

- Authors should specify the statistical method used to calculate Significance (p) in the legend of each table.

- Did the authors used post hoc test to calculate the statistical differences? These information should be added to the statistical analysis section.

Why did the authors not consider the correlation between mERG parameters alteration and arterial hypertension?

- Did the authors observed global differences in amplitude among patients or they obtained responses of approximate equal across the test field for all patients?

- How do they explain that HbA1C>7% correlate with T2DM patients but seems to be not relevant within the same patient group? This finding seems to conflict wit the idea that a good glycemic control reduces the risk of DM-induced retinal alterations.

Minor point:

Typing errors are present in the manuscript   (lines 120, 121, 165, 236, 242)

Author Response

Responses to reviewer #1.

The authors want to thank the reviewer for spending valuable time reviewing this manuscript.

Major points:

  1. The Introduction section need to be improved by checking and reformulating different sentences. In particular, two sentences from line 33 to line 37 need to be clarify and bibliographic entries should be added. In line 48 did the authors want to write “preclude or prelude”? Some general sentences and relatives additional bibliographic entries (such as: PMID: 33065984; 25956512; 32685397) should be added to briefly mention the pathogenesis and genetic implications of DR as well as mostly the potential applications of mERG in the management of the pathology.
  • We have changed lines 33 to 37 to the following:

DR is often not diagnosed until the first signs of vision loss appear. Poorly controlled blood glucose levels can have a serious impact on the development of DR. Patients with T1DM spend many more years of their lives at risk of developing DR than patients with T2DM, since they are exposed to variations in glycemia levels for longer. However, DR has a much greater impact on T2DM, probably because the diagnosis of DM is generally after onset. The patient may have spent several years with high blood glucose levels in a phase of prediabetes or diabetes that remains  unknown for a long time. This will make the detection of DR in patients with T2DM more frequent [3].

  • We have also added the following references:

#7.   Giurdanella G, Lupo G, Gennuso F, Conti F, Furno DL, Mannino G, Anfuso CD, Drago F, Salomone S, Bucolo C. Activation of the VEGF-A/ERK/PLA2 Axis Mediates Early Retinal Endothelial Cell Damage Induced by High Glucose: New Insight from an In Vitro Model of Diabetic Retinopathy. Int J Mol Sci. 2020 Oct 13;21(20):7528. doi: 10.3390/ijms21207528.

#8.   Choudhuri S, Chowdhury IH, Das S, Dutta D, Saha A, Sarkar R, Mandal LK, Mukherjee S, Bhattacharya B. Role of NF-κB activation and VEGF gene polymorphisms in VEGF up regulation in non-proliferative and proliferative diabetic retinopathy. Mol Cell Biochem. 2015 Jul;405(1-2):265-79. doi: 10.1007/s11010-015-2417-z.

#10.  Bian HX, Bian MT, Liu WH, Liu RY, Guo M. Efficiency analysis by mfERG and OCT of intravitreal injection with ranibizumab on diabetic macular edema. Int J Ophthalmol. 2020 Jul 18;13(7):1092-1096. doi: 10.18240/ijo.2020.07.12.

  1. The study seems to be lacking regarding the statistical evaluation of the data. Unlike the authors stated in the statistical methods section (line 125 – 127) the Student's t test is used to compare the means between two groups, whereas ANOVA is used to compare the means among three or more groups but when using one independent variable. Probably the authors used groups with variables. This issue need to be checked. Did the authors checked whether their data are characterized by a normal distribution?We have added included the following sentence to the description of the statistical analysis:
  • The hypothesis of normality in the quantitative variables were tested using the Kolmogorov-Smirnov test.

  1. Authors should specify the statistical method used to calculate Significance (p) in the legend of each table.
  • We have added asterisks in the tables to show them.

  1. Did the authors used post hoc test to calculate the statistical differences? This information should be added to the statistical analysis section.
  • We used a post-hoc quantitative statistical study using Student's t-test or ANOVA. We have added the following sentence to the statistical methods.

In case of three or more samples, we used a post-hoc test. to determine which individual samples were different from one another.

  1. Why did the authors not consider the correlation between mERG parameters alteration and arterial hypertension?
  • We did not consider it because the objective of the study was to compare the differences between patients with T1DM and T2DM, and arterial hypertension affected only 4 patients (11.3% ) of patients with T1DM against 30 patients (84.7%) of patients with T2DM. In any case, we have unpublished data on a possible relationship of T2DM to amplitude alterations, but this relationship could not be evidenced.

  1. Did the authors observed global differences in amplitude among patients or they obtained responses of approximate equal across the test field for all patients?
  • Amplitude was affected in all the patients, both in T1DM and T2DM patients. The decrease in amplitude was not linear, but specific for each patient.

  1. How do they explain that HbA1C>7% correlate with T2DM patients but seems to be not relevant within the same patient group? This finding seems to conflict wit the idea that a good glycemic control reduces the risk of DM-induced retinal alterations.
  • It's true that the levels of HbA1c were more related to the alteration of the implicit time and amplitude in patients with T1DM. This data does not mean that for T2DM patients the two variables were not affected in yhose with elevated hbA1c. The elevated levels of HbA1c affect T2DM patients, but to a lesser extent than T1DM patients. This data is consistent with published results on the greater effect of glycemic variability in T1DM than in T2DM.

Reviewer 2 Report

I enjoyed reviewing this paper. The manuscript is quite interesting. The conclusions are supported by the results. The tables are clear.

However, this reviewer raises some issues that need to be addressed.

1- In conclusions the authors statePatients with type 1 diabetes mellitus, the retina is functionally more sensitive to changes in blood glucose than in patients with type 2 diabetes mellitus. On the contrary, DM duration affects retinal functionality more in both types of diabetes studied.“ These conclusions are very interesting, but they deserve comment and possible explanation. Although in the analysis they are corrected for disease duration, it is well known that type 2 diabetes is, for well-studied pathophysiological reasons, preceded by a long period of prediabetes. Above all, the diagnosis of type-2 diabetes is often late, while that of type 1, due to the more explicit clinical presentation, is precise. It is in fact common to speak of a known duration for type-2 diabetes. Could a long phase of hyperglycemia in a subject with type 2 diabetes not yet diagnosed justify the results of this study?

2- On line 57, the authors describe this investigation as “a prospective case series clinical study”. Actually, as I understand it, this is a cross-sectional study. In any case, it is an observational study. Since recruited diabetic patients have poor glycemic control (HbA1c 7.9%), it cannot be excluded that tight glycemic control may influence the study results. This issue should be addressed by the authors.

3- Although this study assessed multifocal electroretinogram in diabetic subjects without diabetic retinopathy (DR), this complication is the most common in both type 1 and type 2 diabetes. So, early diagnosis of DR is essential to prevent retinal damage leading to blindness. Unfortunately, the diagnosis and staging of diabetic retinopathy encounter several difficulties.  Therefore, particularly in these times of the COVID-19 pandemic, the possibility of diagnosing DR and doing follow-up with telemedicine (1- Diabetes Metab Res Rev. 2019 Mar; 35 (3): e3113. doi: 10.1002 / dmrr.3113 . 2- J Diabetes Res. 2020 Oct 14; 2020: 9036847. doi: 10.1155 / 2020/9036847.) is an important resource, especially in geographic areas where the movement of patients to specialized centers can be long and demanding. This issue as well as above references deserve comment in discussion.

4- The legend of figure 1 should be expanded. JCM is a general medical journal and not a specialist ophthalmology journal. Therefore, the reader should be helped to understand the images.

5- There are some typos in the text. Please correct

Author Response

Responses to reviewer #2.

The authors want to thank the reviewer for spending valuable time reviewing this manuscript.

  1. In conclusions the authors state “Patients with type 1 diabetes mellitus, the retina is functionally more sensitive to changes in blood glucose than in patients with type 2 diabetes mellitus. On the contrary, DM duration affects retinal functionality more in both types of diabetes studied.“ These conclusions are very interesting, but they deserve comment and possible explanation. Although in the analysis they are corrected for disease duration, it is well known that type 2 diabetes is, for well-studied pathophysiological reasons, preceded by a long period of prediabetes. Above all, the diagnosis of type-2 diabetes is often late, while that of type 1, due to the more explicit clinical presentation, is precise. It is in fact common to speak of a known duration for type-2 diabetes. Could a long phase of hyperglycemia in a subject with type 2 diabetes not yet diagnosed justify the results of this study?

  • We have added the following sentence to the Introduction:

DR is often not diagnosed until the first signs of vision loss appear. Poorly controlled blood glucose levels can have a serious impact on the development of DR. Patients with T1DM spend many more years of their lives at risk of developing DR than patients with T2DM, since they are exposed to variations in glycemia levels for longer. However, DR has a much greater impact on T2DM, probably because the diagnosis of DM is generally after onset. The patient may have spent several years with high blood glucose levels in a phase of prediabetes or diabetes that remains  unknown for a long time. This will make the detection of DR in patients with T2DM more frequent [3].

  1. On line 57, the authors describe this investigation as “a prospective case series clinical study”. Actually, as I understand it, this is a cross-sectional study. In any case, it is an observational study. Since recruited diabetic patients have poor glycemic control (HbA1c 7.9%), it cannot be excluded that tight glycemic control may influence the study results. This issue should be addressed by the authors.
  • Sorry, yes, that's an error. We have changed it accordingly.

  1. Although this study assessed multifocal electroretinogram in diabetic subjects without diabetic retinopathy (DR), this complication is the most common in both type 1 and type 2 diabetes. So, early diagnosis of DR is essential to prevent retinal damage leading to blindness. Unfortunately, the diagnosis and staging of diabetic retinopathy encounter severaldifficulties. Therefore, particularly in these times of the COVID-19 pandemic, the possibility of diagnosing DR and doing follow- up with telemedicine (1- Diabetes Metab Res Rev. 2019 Mar; 35 (3): e3113. doi: 10.1002 / dmrr.3113 . 2- J Diabetes Res. 2020 Oct 14; 2020: 9036847. doi: 10.1155 / 2020/9036847.) is an important resource, especially in geographic areas where the movement of patients to specialized centers can be long and demanding. This issue as well as above references deserve comment in discussion.
  • We have added the following to the Discussion:

Finally, in a changing world that has been through the COVID-19 pandemic, the chance of finding new systems for the early detection of  prevalent pathologies such as DR is very important, especially by means of telemedicine [28,29]. This is the case of ERGm, which can be read by ophthalmologists from remote sites.

  • We have also added the following references:

#28. Sasso FC, Pafundi PC, Gelso A, Bono V, Costagliola C, Marfella R, Sardu C, Rinaldi L, Galiero R, Acierno C, de Sio C, Adinolfi LE; NO BLIND Study Group. Telemedicine for screening diabetic retinopathy: The NO BLIND Italian multicenter study. Diabetes Metab Res Rev. 2019 Mar;35(3):e3113. doi: 10.1002/dmrr.3113.

#29.  Galiero R, Pafundi PC, Nevola R, Rinaldi L, Acierno C, Caturano A, Salvatore T, Adinolfi LE, Costagliola C, Sasso FC. The Importance of Telemedicine during COVID-19 Pandemic: A Focus on Diabetic Retinopathy. J Diabetes Res. 2020 Oct 14;2020:9036847. doi: 10.1155/2020/9036847

  1. The legend of figure 1 should be expanded. JCM is a general medical journal and not a specialist ophthalmology journal. Therefore, the reader should be helped to understand the images.
  • We have added the following figure legend:

Figure 1.  Diagram of the three central rings studied. On the right, the position of the three central rings of the mERG with respect to the eye fundus. On the left, the waves of the five rings of the ERGm, of which we only use the three central ones. This image shows the implicit time (triangle with the vertex up) and the amplitude (triangle with the vertex down)”

  1. There are some typos in the text. Please correct
  • We have corrected any errors accordingly.

Round 2

Reviewer 2 Report

No further comments.